

# Antimicrobial and antioxidant potential from *Piper marginatum* roots

Adson Soares da Silva[1], Janete Magali da Silva[2] and Clécio Souza Ramos[3]

[1] Department of Chemistry, UFRPE, Recife, PE, Brasil
[2] Antibiotic, Federal University of Pernambuco, Recife, PE, Brasil
[3] Department of Chemistry, Rural Federal University of Pernambuco, Recife, PE, Brasil

## ABSTRACT

This is the first report of the antimicrobial and antioxidant potential of extract from *Piper marginatum* roots. The extract showed highest antioxidant activity with an $EC_{50}$ of $47.3 \pm 0.80\ \mu g/mL$ and a total phenolic content of $42.7 \pm 1.10$ mg GAE/g. The extract exhibited strong antimicrobial activity with minimum inhibitory concentrations of 250, 250 and 125 µg/mL for the bacteria *Bacillus subtilis*, *Staphylococcus aureus* and fungus *Mycobacterium smegmatis*. Antimicrobial activity was attributed to (*E,E*)-*N*-Isobutyl - 2,4-octadienamide and (*E,E*)-*N*-Isobutyl-2,4-decadienamide amides isolated as major compounds of the roots. Structural elucidation of the two amides was determined based on the interpretation of their IR, UV, MS, $^1$H and $^{13}$C NMR spectra. The results contribute significantly to the development of a herbal remedies based on *P. marginatum* roots.

## INTRODUCTION

Natural products have been considered an excellent source of new molecules as a prototype in the discovery of new drugs (*Silva et al., 2021*). In the period of 30 years (1981 to 2014), 43.5% of the drugs approved worldwide for the treatment of infections caused by bacteria, fungi, parasites and viruses were obtained from natural products (*Cragg & Newman, 2013*). Taxol is one of the remarkable natural anticancer drugs, firstly extracted from the *Taxus brevifolia* plant in 1971 and has been useful in the management of many cancers (*Teibo et al., 2023*). Artemisinin is another example of a natural source drug; a sesquiterpene extracted from *Artemisia annua* and widely used in Chinese medicine for the treatment of malaria (*Weathers, 2023*).

As part of our systematic chemical and biological study with plant species of the Piperaceae family (*Da Silva et al., 2018*; *Rocha et al., 2016*; *Barbosa et al., 2012*), the present work was directed to evaluate the antimicrobial and antioxidant activity as well as the structural determination of the specialized metabolites of extracts from *Piper marginatum* roots. *P. marginatum*, popularly known as Pimenta do Mato, Malvaísco or Capeba Cheirosa, is a plant used in folk medicine to relieve stomach pains, as a diuretic, for snake bites and as a carminative (*Brú & Guzman, 2016*). It has exhibited diverse activities including cytotoxic, larvicidal, antimicrobial, insecticidal, antioxidant, repellent, anti-alimentary and

Corresponding author
Clécio Souza Ramos,
clecio.ufrpe@gmail.com

phytotoxic (*Macêdo et al., 2020*; *Brú & Guzman, 2016*). Previous chemical studies have revealed that the *P. marginatum* leaves accumulate prenylated benzoic acids, amides and flavonoids while essential oils were rich in phenylpropanoids (*De Oliveira, Silva & Ramos, 2022*; *Santos Ayres et al., 2022*). Previous studies revealed that dichloromethane extract and essential oil from *P. marginatum* leaves rich in phenylpropanoids exhibited antimicrobial activity against pathogens of clinical importance (*Bezerra & Ramos, 2021*). Despite wide use of *P. marginatum* in folk medicine and its variety of biological activities; phytochemical and biological studies with extracts from the roots of *P. marginatum* are still rare (*Takeara et al., 2017*; *Bay-Hurtado et al., 2016*; *Sequeda-Castañeda et al., 2015*).

## MATERIALS & METHODS

### Chemicals

Folin-Ciocalteu's phenol reagent, potassium persulfate and 6- hydroxy-2,5,7,8-tetramethylchroman-2-carboxylic acid (Trolox) were obtained from Sigma-Aldrich (Waltham, MA, USA), and 2,2- azinobis 3-ethylbenzothiozoline-6-sulfonic acid (ABTS) was supplied by Fluka Chemie (Buchs, Switzerland).

### Plant

The roots of *P. marginatum* were collected in an Atlantic forest fragment located on the Campus of the Federal Rural University of Pernambuco (UFRPE), Recife, Pernambuco, Brazil (8°00′53.4″S 34°57′05.5″W). The botanical material was identified by Prof. Dr. Margarethe Ferreira de Sales from the Department of Biology at UFRPE. A specimen of the collected botanical material was deposited at the Vasconcelos Sobrinho Herbarium of the Department of Biology at UFRPE under number 45870.

### Extraction and compounds isolation

*P. marginatum* roots (500 g) were dried in a circulating air oven for a period of 48 h at 50 °C and the dried material was milled to a fine powder in a Macsalab mill (400 g). The dried root was extracted by maceration with n-butanol ($3 \times 400$ mL) at room temperature for 48 h and concentrated in a vacuum to yield crude extract (8 g). Part of the extract (4.0 g) was subjected to fractionation on a Sephadex LH-20 (Amersham Biosciences, Amersham, UK) using methanol as eluent, yielding 84 fractions. Fractions 23 to 31 were pooled (350 mg) and further fractionated with silica gel (Merck 230-400 mesh) using hexane with increasing amounts of ethyl acetate (EtOAc) as the eluent, yielding 40 fractions with isolation of compounds 1 (20.4 mg) and 2 (18.6 mg).

### HPLC analyses

HPLC analyses of extracts and the pure compound were performed using a Shimadzu LC10 instrument (Tokyo, Japan) with an SPD-M20A diode array detector using a $C_{18}$ column (250 mm, 4.6 mm, 5 $\mu$M, (Supelco, Bellefonte, PA, USA) eluted in a gradient mode starting with $CH_3OH:H_2O$ (3:7) for 10 min, raising to 100% of $CH_3OH$ (HPLC grade, Merck-Brazil, Millipore, Burlington, MA, USA) in 40 min and flow rate of 1.0 mL/min. The injection volume was 20 $\mu$L of solution 1 mg/mL in methanol.

## Structure

$^{1}$H and $^{13}$C NMR spectra of 1 and 2 compounds were acquired in $CDCl_3$ on Bruker DPX-300 (Bruker, BioSpin GmbH Silberstreifen 476287 Rheinstetten Germany; 300 MHz and 75 MHz, respectively) spectrometer with pulsed field gradient and signals referenced to the residual solvent signals ($CDCl_3$, at $\delta$ H 7.26 and $\delta$ C 77.0 ppm, 99% purity Aldrich, St. Louis, MO, USA). GC-MS analyses (60−260 °C at 3 °C/min. heating rate) were carried out in a Varian 431-GC coupled to a Varian 220-MS instrument using (Palo Alto, CA, USA) fused-silica capillary column (30 m × 0.25 mm i.d. × 0.25 μm) coated with DB-5. MS spectra were obtained using electron impact at 70 eV with a scan interval of 0.5 s and fragments from 40 to 550 Da. The injection volume was 1.0 μL of solution 1 mg/mL in methanol. IR spectra were obtained on a Perkin Elmer Nicolet 1750 using KBr disk (Palo Alto, CA, USA). The analyses UV-Vis were carried out Agilent 8453 spectrophotometer (Santa Clara, CA, USA), in the interval from 190 to 600 nm, using 10-mm quartz cuvettes and solution (0.5 mg/mL) in methanol.

## Total phenolic content

The total phenolic content of the samples was determined using the Folin-Ciocalteu reagent (Sigma-Aldrich, St. Louis, MO, USA) according to the method previously reported with slight modifications of gallic acid as a standard phenolic compound (*Slinkard & Singleton, 1977*). Appropriate amounts of each sample (500 μL; 50 μg/mL) were diluted in a volumetric flask with distilled water (3 mL). The Folin-Ciocalteu reagent (100 μL) was added and the contents of the flask were thoroughly mixed. After 3 min, $Na_2CO_3$ (15%, 300 μL) was added and the mixture was completed with distilled water (5 mL) and allowed to stand for 2 h in an ultrasonic bath. The absorbance was measured at 760 nm in a spectrophotometer. The total amount of phenolic compounds was determined in micrograms of gallic acid equivalents, using the equation obtained from the standard gallic acid graph.

## ABTS$^{\bullet+}$ radical scavenging assay

The method previously reported with slight modifications was adopted for ABTS (2,2′-azino-bis-(3-ethylbenzothiazoline-6-sulfonate, 98% purity) assay (*Re et al., 1999*). ABTS$^{\bullet+}$ was generated by reacting ABTS solution (7 mM) with potassium persulphate (140 mM, final concentration) for 16 h in the dark at room temperature. Then, the ABTS$^{\bullet+}$ solution was diluted with ethanol to obtain absorbance of 0.70 ($\pm$ 0.02) at 734 nm and further equilibrated at 30 °C. The samples were diluted in ethanol at a concentration of 1 mg/mL, and the ABTS$^{\bullet+}$ solution was added to both samples to obtain concentrations of 1.0 to 100.0 μg/mL. The absorbance of the reaction mixture was measured at 734 nm after reaction at room temperature for 10 min. Trolox (97% purity) was used as positive control. The capability of scavenging the ABTS$^{\bullet+}$ radical was calculated using the following equation: ABTS$^{\bullet+}$ scavenging effect (%) = [(A0 − A1 / A0) ×100].

Where A0 is the initial concentration of ABTS$^{\bullet+}$ and A1 is the absorbance of the remaining concentration of ABTS$^{\bullet+}$ in the presence of the sample.

### In vitro antimicrobial activity

The antimicrobial potential of extract and compounds was evaluated against the gram-positive bacteria *Staphylococcus aureus*, *S. aureus MRS*, *Bacillus subtilis* and the gram-negative bacteria *Escherichia coli*, *Klebsiella pneumoniae*, *Pseudomonas aeruginosa* as well as the fungi *Candida albicans*, *Mycobacterium smegmatis* and *Malassezia furfur*. The bacteria and fungi came from the collection of microorganisms from the Antibiotics Department of the Federal University of Pernambuco. The suspension of microorganisms was standardized by the turbidity equivalent to a 0.5 tube on the McFarland scale in distilled water, corresponding to a concentration of approximately 108 CFU/ml for bacteria and 107 CFU/ml for fungi (*Ramos et al., 2022*; *Bezerra & Ramos, 2021*).

### Determination of the minimum inhibitory concentration (MIC)

MIC was performed using the microdilution technique in 96-well multiplates. The culture media used were Sabourand Agar (for fungus) and Muelle-Hinton Agar (for bacteria). Metronidazole (2.5 µg/ml) and Fluconazole (2.5 µg/ml) were used as a positive control, while ethyl alcohol was used as a negative control. Analyses were performed in triplicate and the microplates were cultured at 37 °C for 18–24 h for bacteria and 30 °C for 48–72 h for the fungus. After the culture period, the microplates were developed with the addition of 10 µL of 0.01% resazurin solution and incubated for 3 h. The MIC was defined as the lowest concentration of the sample that inhibited the growth of the microorganisms (*Ramos et al., 2022*; *Bezerra & Ramos, 2021*).

## RESULT & DISCUSSION

The chemical profile obtained by HPLC of the butanolic extract from *P. marginatum* roots showed two major peaks (Fig. 1).

The extracts were submitted to purification steps by chromatographic methods enabling isolation of two compounds 1 and 2. The UV, IR, $^{13}$C and $^1$H NMR spectra for compounds 1 and 2 were identical, indicating that the chemical structures of these compounds are similar. $^{13}$C and $^1$H NMR spectra of compound 1 showed a set of signals characteristic of alkylamides (Table 1). Mass spectrum for compound 1 showed a molecular ion peak of *m/z* = 195 Da, and the mass spectrum of compound 2 showed molecular ion peak of *m/z* = 223 according to the molecular formulas $C_{12}H_{21}NO$ and $C_{14}H_{25}NO$, respectively. Based on interpretation of IR, MS, $^{13}$C and $^1$H NMR spectral data, compounds 1 and 2 were determined to be (*E,E*)-*N*-Isobutyl-2,4-octadienamide and (*E,E*)-*N*-Isobutyl-2,4-decadienamide amides, respectively.

The (*E,E*)-*N*-Isobutyl-2,4-octadienamide and (*E,E*)-*N*-Isobutyl-2,4-decadienamide amides previously isolated from *Piper sarmentosum* aerial parts, *P. nigrum* seeds and *Cissampelos glaberrima* roots showed insecticidal activity (*Siddiqui et al., 2005*; *Stöhr, Xiao & Bauer, 1999*; *Rosario, Da Silva & Parente, 1996*). This is the first report of *E,E*)-*N*-Isobutyl-2,4-decadienamideamide in *P. marginatum* tissues. The (*E,E*)-*N*-Isobutyl-2,4-octadienamide has been previously of aqueous ethanolic extract from *P. marginatum* roots (*De Oliveira Santos & De Oliveira Chaves, 1999*).

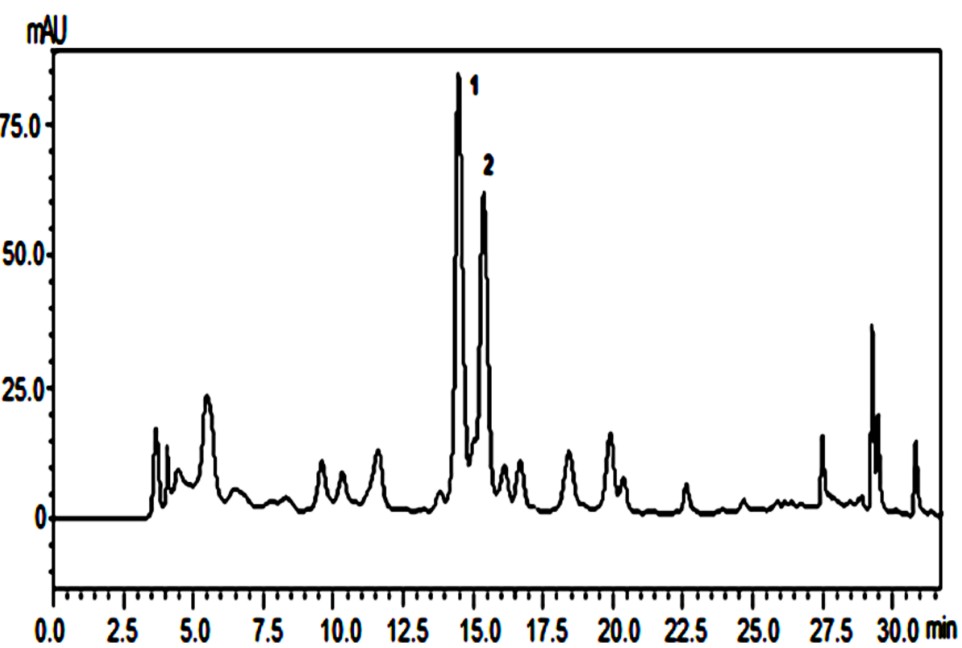

**Figure 1** Chemical profile of butanolic extract from *P. marginatum* roots.

**Table 1** ¹H and ¹³C NMR spectral data of the compounds 1 and 2.

| H/C H/C | (E,E)-N-Isobutyl-2,4-octadienamide (1) | | (E,E)-N-Isobutyl-2,4-decadienamide (2) | |
|---|---|---|---|---|
| | $\delta_H$ | $\delta_C$ | $\delta_H$ | $\delta_C$ |
| 1 | – | 166.77 | – | 166.40 |
| 2 | 5.75 (d, 1H, $J = 14.9$ Hz) | 143.30 | 5.70 (d,1H, $J = 15.2$ Hz) | 143.31 |
| 3 | 6.09 (m, 2H) | 141.68 | 6.12 (m. 2H) | 141.38 |
| 4 | 6.09 (m, 2H) | 128.77 | 6.12 (m, 2H) | 128.19 |
| 5 | 7.18 (dd, 1H, $J = 14.9$ and 9.6 Hz) | 122.17 | 7.20 (dd, 1H, $J = 15.2$ and 9.9 Hz) | 121.67 |
| 6 | 2.12 (q, 2H, $J = 7.0$ Hz) | 35.38 | 2.14 (q, 2H, $J = 5.5$ Hz) | 32.93 |
| 7 | 1.44 (m, 2H) | 29.05 | 1.79 (m, 2H) | 31.38 |
| 8 | 0.91 (m, 3H) | 14.08 | 1.29 (m, 2H) | 28.64 |
| 9 | – | – | 1.40 (m, 2H) | 28.49 |
| 10 | – | – | 0.91 (t, 3H) | 14.02 |
| 1' | 3.16 (d, 2H, $J = 6.7$) | 47.32 | 3.16 (d, 2H, $J = 6.7$ Hz) | 46.93 |
| 2' | 1.81 (m, 1H) | 22.43 | 1.80 (m, 2H) | 22.48 |
| 3' | 0.90 (d, 6H, $J = 7.0$) | 20.54 | 0.90 (d, 6H, $J = 6.7$ Hz) | 20.13 |
| 4' | 0.90 (d, 6H, $J = 7.0$) | 20.54 | 0.90(d, 6H, $J = 6.7$ Hz) | 20.13 |
| N-H | 5.51(s, 1H) | – | 5.45 (s,1H) | – |

**Table 2** Zone of inhibition and MIC of samples from *P. marginatum* roots.

| Microorganisms | Zone of *inhibition* (mm) | MIC (μg/ml) | | |
|---|---|---|---|---|
| | Extract | Extract | Compound 1 | Compound 2 |
| *B. subtilis* | 11 | 250 | 250 | 250 |
| *S. aureus* | 22 | 250 | 250 | 250 |
| *S. aureus MRS* | 0 | – | – | – |
| *K. pneumoniae* | 0 | – | – | – |
| *E. coli* | 0 | – | – | – |
| *P. aeruginosa* | 0 | – | – | – |
| *M. furfur* | 0 | – | – | – |
| *C. albincas* | 0 | – | – | – |
| *M. smegmatis* | 15 | 125 | 125 | 125 |

## Antimicrobial activity

The crude extract from *P. marginatum* roots was tested against nine microorganisms and showed an inhibition zone for two bacteria and one fungus, indicating a selective antimicrobial activity for these microorganisms (Table 2).

The minimum inhibitory concentrations (MIC) of the extract for the bacteria *B. subtilis*, *S. aureus* and fungus *M. smegmatis* were determined with values of 250, 250 and 125 μg/mL, respectively (Table 2). MIC values $\leq$ 500 μg/ml are considered strong inhibitors; MIC values from 600 to 1,500 μg/ml are considered moderate inhibitors; values 1,500 μg/ml are considered weak inhibitors (*Aligiannis et al., 2001*). The (*E,E*)-*N*-Isobutyl-2,4-octadienamide and (*E,E*)-*N*-Isobutyl-2,4-decadienamide amides showed strong antimicrobial activity against *B. subtilis*, *S. aureus* and *M. smegmatis* with value of 250 μg/mL for bacteria and 125 μg/mL for fungus, indicating that the amides are responsible for activity of butanolic extract from *P. marginatum* roots. A previous study revealed that essential oil from leaves of *P. marginatum* rich in phenylpropanoids showed antimicrobial activity against bacteria *B. subtilis, S. aureus* and fungus *M. smegmatis* with MIC values of 250, 2,500, 1,250 μg/mL, respectively (*Ramos et al., 2022*; *De Oliveira, Silva & Ramos, 2022*). The leaf extract showed activity against the bacteria *S. aureus* with a MIC of 1,250 μg/mL (*Bezerra & Ramos, 2021*).

## Antioxidant activity and total phenolic content

The results obtained in the determination of total phenolics were expressed in milligrams equivalent of gallic acid per gram of extract (mg GAE/g, Table 3).

The crude root extract exhibited a total phenolic content of 42.7 $\pm$ 1.10 mg GAE/g, indicating that in addition to amides, the root extract also accumulates phenolic compounds in moderate amounts. The butanolic extract from *P. marginatum* roots showed the highest activity with an $EC_{50}$ = 47.3 $\pm$ 0.80 μg/mL. $ABTS^{·+}$ radical scavenging activity of roots was associated with the phenolic content, considering that $EC_{50}$ = for the 383.0 $\pm$ 5.19 and 340.4 $\pm$ 4.81 μg/mL, respectively, considering that values approaching 500 μg/mL do not exhibit good scavenging capacity. To our knowledge, this is the first report of the antioxidant activity for the extract and their major constituents from the roots of

**Table 3  Total phenolic content butanolic and ABTS radical scavenging activity of extract and amides from *P. marginatum* roots.**

| Samples | ABTS EC$_{50}$ µg/mL | Total phenolic [a]mg GAE/g |
|---|---|---|
| Extract | 47.3 ± 0,80 | 42.7 ± 1,10 |
| (*E,E*)-*N*-Isobutyl-2,4-octadienamide | 383.0 ± 5.19 | – |
| (*E,E*)-*N*-Isobutyl-2,4-decadienamide | 340.4 ± 4.81 | – |
| Trolox | 2.8 ± 0,00 | – |

**Notes.**
[a]mg GAE/g = mg of gallic acid per g of extract; EC$_{50}$ = effective concentrations.

*P. marginatum*. Previous studies with essential oil of *P. marginatum* showed significant antioxidant activity with a DPPH IC$_{50}$ value between 1,200 and 1,500 µg/ml while the control ascorbic acid showed a DPPH IC$_{50}$ value of 1,000 µg/ml (*Brú & Guzman, 2016*). Essential oil from the roots of *P. marginatum* exhibited antioxidant activity with a DPPH IC$_{50}$ value of 75,300 µg /L while the *Ginkgo biloba* used as reference exhibited a DPPH IC$_{50}$ value of 46.9 mg/L (*Bay-Hurtado et al., 2016*). *Piper umbellata* has been reported to be a potent antioxidant attributed to its major constituent, 4-nerolidylcatechol (*Cordeiro et al., 2013*). The total phenolic concentration of the extract from the leaves of *P. umbellata* was of 148 mg GAE/g while antioxidant activity exhibited an EC$_{50}$ of 120.1 µg/mL based on ABTS (*Ramos et al., 2012*).

## CONCLUSION

This is the first report of the antimicrobial and antioxidant potential of extract from *P. marginatum* roots. The results revealed that the plant exhibits a selective antimicrobial activity against the bacteria *B. subtilis*, *S. aureus* and fungus *M. smegmatis* attributed to (*E,E*)-*N*-Isobutyl -2,4-octadienamide and (*E,E*)-*N*-Isobutyl-2,4-decadienamide amides. The extract also exhibited antioxidant activity which was attributed to the phenolic content present in the plant. The results contribute significantly to the chemical and biological knowledge of *P. marginatum*, a plant widely used in folk medicine.

## ACKNOWLEDGEMENTS

This work was funded by grants from FACEPE. ASS thanks CAPES for providing a scholarship. The authors are indebted to the Centro de Apoio a Pesquisa (CENAPESQ), UFRPE, for the laboratory facilities.

### Funding

This work was funded by grants from FACEPE. Adson Soares da Silva received a scholarship from CAPES. The funders had no role in study design, data collection and analysis, decision to publish, or preparation of the manuscript.

## Grant Disclosures

The following grant information was disclosed by the authors:
FACEPE.
CAPES.

## Competing Interests

The authors declare there are no competing interests.

## Author Contributions

- Adson Soares da Silva conceived and designed the experiments, performed the experiments, analyzed the data, performed the computation work, prepared figures and/or tables, authored or reviewed drafts of the article, and approved the final draft.
- Janete Magali da Silva conceived and designed the experiments, performed the experiments, authored or reviewed drafts of the article, and approved the final draft.
- Clécio Souza Ramos conceived and designed the experiments, performed the experiments, analyzed the data, prepared figures and/or tables, authored or reviewed drafts of the article, and approved the final draft.

## Data Availability

The raw measurements are available in the Supplemental Files.

## Supplemental Information

Supplemental information for this article can be found online at http://dx.doi.org/10.7717/peerj-ochem.8#supplemental-information.

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
