# Peer review of "Antimicrobial and antioxidant potential from *Piper marginatum* roots"

_PeerJ Organic Chemistry, doi:10.7717/peerj-ochem.8_

## Round 0.1 · original submission · Major Revisions

Dear prof. Ramos,

Thank you again for your manuscript submission: “Antimicrobial and antioxidant potential of alkylamides isolated from Piper marginatum roots”. Your manuscript has now been reviewed by experts in the field.

Our impression is that the above-mentioned manuscript might indeed become suitable for publication in PeerJ Organic Chemistry after major revision. The comments of the referees are included at the bottom of this letter.

A revised version of your manuscript that takes into account the comments of the referees will be reconsidered for publication.

Please note that submitting a revision of your manuscript does not guarantee eventual acceptance and that your revision will be subject to re-review by the referees before a decision is rendered.

Editor comments:

The introduction could start with line 39: “Piper marginatum, popularly known as...” All the sentences before line 39 could be deleted.
Lines 47-48: What did the authors mean by “There are many reports comparing the leaves with phytochemical studies from extracts.”? Would have these studies made a correlation between biological activities/use in folk medicine and the phytochemical profile of extracts?
Materials and methods:
- Please, provide the GPS coordinates
- p. 77: Please, correct this subtitle. Suggestion: Extraction and compounds isolation
- line 78: how did the authors dry the roots before the extraction:? Did they use air drying or circulating-air oven stove?
- line 88: why was the purpose of the HPLC analyses?
- line 92: compounds 1 and 2 must be bolded
- line 95: Please, change the subtitle “Characterization chemical” to “Structure –
- Lines 110 and 113: please, change ml to mL
- line 165: based on which literature criteria the authors classified the antimicrobial activity as “strong”?
- Table 1: the J values of H5 are missing for compounds 1 and 2.
- Description of the antimicrobial assays was not given in Materials and methods.

Please do not hesitate to contact us if you have any questions regarding the
revision of your manuscript or if you need more time. We look forward to
hearing from you soon.
Best regards,
Antônio E. Miller Crotti
Associate Editor
PeerJ Organic Chemistry

·

Basic reporting

In the manuscript, the authors studied the antimicrobial and antioxidant activity of P. marginatum, common in some regions of Brazil. The introduction is concise, but the experimental section is incomplete, and results and discussion must increase in quality of the discussions, some data must be provided to increase the quality of the manuscript. The manuscript has merits, nevertheless, some points must be added and reviewed. For these reasons, and for the reasons described below I suggest the manuscript can be published after major revisions.

Experimental design

In the introduction, the authors develop an interesting discussion about the importance of the Piper genus, the importance of secondary metabolites of Piper species, and introduce the research problem, the use of root extracts of P. marginatum on antimicrobial and antioxidant activities. The introduction is well conducted, and the references are actualized.
The materials and methods section seems concise, but must increase in quality, especially in terms of experimental data detailing. Some data must be provided as discussed below. My great criticism is about the antimicrobial activity, which is not described in the experimental section. Please insert complete data (strains, methods...) well-referenced in the experimental section.
- On p. 7 l. 53-56, please insert the purity of the Trolox and ABTS used in the experiments.
- On p. 7 l. 60, please insert the geographical coordinates where the plant material was obtained.
- On p. 7 l.69 please insert the brand and origin of the solvent used. The same information must be provided for the Sephadex, methanol, silica gel, hexane, and ethyl acetate (p.8 l.71-73).
- On p. 8 l.73, please change “EtOAc” to “ethyl acetate (EtOAc)”.
- On p. 8 l.78-80, please insert the origin of the HPLC equipment, the column, and the purity, brand, and origin of the MeOH.
- On p. 8 l.83 please insert the purity, brand, and origin of the deuterated solvent.
- On p. 8 l.84, please insert the origin of the NMR equipment.
- On p. 8 l.86, please insert the model and origin of the MS equipment. In addition, some information about the equipment conditions is required, please complete the conditions.
- On p. 8 l.86, please insert the origin of the FT-IR equipment. In addition, some information about the equipment conditions is already required, please insert the complete conditions for analysis. The same observations must be followed by UV-Vis analysis.
- On p. 9 l.96 please insert the brand, purity, and origin of the Folin-Ciocateu reagent used. On the same page, l.99, please insert the data of the spectrophotometer used in the Total Phenolic Content analysis.
- On p. 9 l.106 please insert the brand, origin, and purity of the reagents. In addition, insert the data of the equipment used in ABTS analysis.

Validity of the findings

The results and discussion must increase in quality to be published, some data must be better organized and better discussed, as described below.
- Figure 1, cited on p. 10 l.120 is low quality, please increase the quality of the chromatogram. In addition, the authors do not cite efforts to include in the work the other compounds observed on the chromatogram. Were the other compounds analyzed?
- In table 1 cited on p. 10 l.125, some data present on the table must be revised: please change “,” to “.” in all cases on chemical shifts and coupling constants; all chemical shifts must be expressed with the same number of significant numbers. On compound 1, some coupling constants (J) are not expressed, e.g. H5 for (1) and H5 for (2).
- On p. 10 l. 126-127, “m/z” must be expressed in italics. In addition, please insert the MS spectrum of compounds 1 and 2 in the supplementary file.
- The antimicrobial activity section (p.11) must increase quality substantially, some data is described in the text, but I strongly suggest including a table exhibiting the results of antimicrobial activity.
- On p. 11 l.147, the authors described the antimicrobial activity as “strong”, why? Please provide a pattern to affirm it for isolated compounds and extracts. The comparative analysis of antimicrobial activity must be better discussed.
- On p. 11, l.160 the authors cited in Table 3 data of ABTS and EC50. However, Table 3 is the same as Table 2. The authors must provide the correct table. I suggest organizing Tables 2 and 3 in an only table.
- The antioxidant activity section must increase in quality by a discussion comparing the results with the literature. No comparisons with other experiments in the literature are established. I suggest the authors compare it with other fractions of Piper genus, including works of P. marginatum available in the literature (e.g. DOI:10.5902/2179460X21803).

Additional comments

The references must be revised mainly in details of formatting, uppercase and lowercase letters in the titles or names of journals must be standardized, and abbreviations and full names of the journals must be standardized.

Reviewer 2 ·

Basic reporting

The manuscript “Antimicrobial and antioxidant potential of alkylamides isolated from Piper marginatum roots” details about antimicrobial and antioxidant potential of extract from P. marginatum roots. In addition, three alkylamides have been isolated as the main compounds of such roots.
Authors attributed the extract's antimicrobial and antioxidant activities to the presence of alkylamides, but the results are not enough to confirm that.
I cannot recommend the publication of the study in its present form.
Below are some suggestions for authors for further resubmission.

Experimental design

The study was well designed regarding the experimental section, but lacks information on how the experiments were done and how the results were achieved.
The study falls within Scope of the Journal.

Validity of the findings

The manuscript details about antimicrobial and antioxidant potential of extract from P. marginatum roots.
Authors attributed the extract's antimicrobial and antioxidant activities to the presence of alkylamides, but the results are not enough to confirm that. The chemical characterization should be improved.

Additional comments

All text needs improvement in language and reorganization.
The abstract should be rewritten for the organization of the resented information. I suggest that all bioactivities displayed by roots extract are presented at the start. For this, the antioxidant results should be put at to start of the section. After this, the isolation and chemical characterization of three alkylamides should be discussed.
Keywords: should be reformulated. They must be representative of the study.
The introduction is too short. Authors could provide, for instance, more details about medicinal plants' contribution to developing new herbal medicines. The use of “secondary metabolite” must be carefully evaluated as it has been substituted by “specialized metabolites”
The aims of the study are depicted in lines 46-48. I recommend a complete reformulation, as only three secondary metabolites have been isolated. The term “elucidation” should be replaced by structural determination.
Materials and methods lack too much information:
Chemicals: information about solvents precedence
Plant: information about geographic coordinates (GPS signal) of the place of collect of the roots.
Obtaining and isolation of extract: the achievement of compound 3 had not been mentioned.
HPLC analysis: more details about the equipment and software should be provided.
Characterization chemical: More information about EI-MS analysis should be provided
The methodology for the evaluation of antimicrobial activity was not provided
Results and Discussion: this section needs to sound scientific. It requires a complete reformulation. How has compound 3 been isolated and identified?
The authors mentioned that all spectroscopic data of compounds 1 and 2 are identical (line 123). Probably, they are similar, or the compounds 1 and 2 are the same.
How have the stereochemistries been suggested?
The reference list should be adequate for Instructions to the authors.

Reviewer 3 ·

Basic reporting

The work has scientific merit and a final english revision is required.

Materials and Methods section
- Provide the complete name of the studied species with the botanist.
- “Obtaining and isolation of extract compounds” or “Obtaining and fractionating of extract"
- specify which butanol (eg. n-butanol)

Results and Discussion section
- If spectral data are identic the compounds would be the same. x "The spectra are similar...."
- One would expect that an isolated compound would be more potent than a extract, since the isolated compound is “diluted in the extract”. Considering the same activity (extract x isolated compound), a proper discussion explaining the results (synergic effect or not) is recommended.

Experimental design

The statements of “Strong selective antimicrobial activity” seems too much... It would be preferreble if authors show absence of toxic effects in non-target cells if they claim the "selectivity" or remove the statement.

Validity of the findings

No comment

---

## Round 0.2 · Minor Revisions

The authors have made most of the changes that were suggested by the reviewers and the revised version is much more clear and concise. However, the reviewers considered that a few minor revisions are still needed, especially in the English language. Please, make a revision in the English language and consider the changes suggested by the reviewer before submitting a R2 version of the manuscript.

·

Basic reporting

The authors submitted the major changes suggested by the reviewer, please insert the final changes as described below. I suggest the manuscript can be published after minor revisions.

Experimental design

Please, insert the proper references to the methods of In vitro antimicrobial activity and Determination of MIC described on p.6-7.

Validity of the findings

In the first review, it was solicited to change “,” to “.” for the chemical shifts and coupling constants, but in some cases, the mistake remains. In addition, some coupling constants remain (J) a not expressed, e.g. H5 for (1) and H5 for (2).
On p. 10 l. 126-127, “m/z” must be expressed in italics. In addition, please insert the MS spectrum of compounds 1 and 2 in the supplementary file. The correction was not done.

Additional comments

The references must be again, and details of formatting must be standardized. E.g. Aligiannis 2001, Macedo 2020, Re 1999, Rosario 1996, Siddiqui 2005, Slinkard 1977.

---

## Round 0.3 · accepted · Accept

The authors have replied to all the reviewer's comments and made all the changes that were recommended. I am happy with the current version, which I consider ready for publication.